# Forgiveness and Reconciliation in Palliative Care: The Gap between the Psychological and Moral Approaches

**Carlo Leget**

Department of Care Ethics, University of Humanistic Studies, 3512 HD Utrecht, The Netherlands;
C.Leget@UvH.nl

**Abstract:** Forgiveness is an important theme in end-of-life care in all spiritual and religious traditions, although it is framed differently. Looking at research on forgiveness in palliative care literature from the last two decades, it is clear that forgiveness is predominantly carried out from a psychological perspective. According to this approach, forgiveness is seen as something that can be managed and taught in order to reduce stress and promote health. There is no doubt that this approach has its merits and is useful for dealing with guilt from the individual perspective of one's own psychological health. From a moral perspective, however, forgiveness is more than dealing with personal feelings of guilt. In order to show the differences and gaps between the psychological and moral perspectives on forgiveness, I discuss the work of the German philosopher Svenja Flaßpöhler. I show that, from a moral perspective, forgiveness can neither be managed or taught, nor seen as a form of understanding, loving, or forgetting. As a conclusion, I formulate some recommendations for future research on forgiveness, distinguishing between the psychological and moral perspectives on forgiveness.

**Keywords:** forgiveness; guilt; reconciliation; palliative care; end of life; moral; psychological

## 1. Introduction: Defining the Problem

Forgiveness, reconciliation, and the end of life are intrinsically connected. At the close of life, many people (in many different ways) look back upon their lives. Coming to a narrative closure implies contemplating and evaluating what has happened during one's life time. For many people, this includes guilt and unfinished business.

What seems to be a universal human feature when confronted with finitude and mortality is also part of religious traditions and philosophies of life. In their overview of the spiritual importance of end-of-life care among five major faiths of the United Kingdom, Choudry et al. (2018) give an overview of the most important characteristics of Christianity, Islam, Judaism, Hinduism, and Sikhism. Showing how these religions provide guidance on how to live purposefully (and how life is seen as a place of moral and spiritual development), it is clear that the close of life is related to such themes as forgiveness and reconciliation. This goes all the more for those traditions in which the end of life is connected to passing from the physical world to beyond—a passage mostly related with an evaluation of life on Earth and a punishment or reward in afterlife.

Since palliative care is a multidimensional approach that deals with quality of life of patients faced with life-threatening illnesses and addresses psychosocial and spiritual problems, one would expect ample attention to be paid to forgiveness and reconciliation here. Although in the practice of palliative care, these themes are undeniably present in everyday care, research on palliative care shows a different picture. Despite the fact that there are some empirical studies on different aspects of the subject (forgiveness and reconciliation are touched upon in a number of instruments used in

palliative care), the theme is less present and researched than one might expect. Analyzing these studies, one cannot but conclude that, in the way the subject is discussed, more attention is paid to the psychological dimension of these phenomena than the spiritual and/or religious dimension. This raises serious concerns for the quality of palliative care.

I wish to address these concerns and ask some critical questions about the way forgiveness and reconciliation are discussed and researched in palliative care. I will formulate a clear diagnosis and propose some modest but clear directions for further research. In order to prepare my analysis of research on forgiveness and reconciliation at the end of life, I will begin by introducing the context (research on spiritual care in palliative care) and make two important observations on this field of studies. These observations set the scene for what follows. Subsequently, I will discuss some recent research on the subject and show how these works share an implicit psychological approach to forgiveness and reconciliation. In order to understand the limits of this implicit approach, I will discuss an influential author in the field of psychology and contrast his view on forgiveness with a moral approach to forgiveness. This moral approach to forgiveness is more difficult to reconcile or combine with a spiritual approach than one might expect. In order to show the complexity of a moral approach, I will use the recent work of the German philosopher Svenja Flaßpöhler. Because palliative care is open to patients and families from any spiritual, religious, or non-religious tradition whatsoever, I will formulate my analysis and critique in philosophical terms. I will end with some concluding remarks and questions for further research.

## 2. The Context: Spiritual Care in Palliative Care

What has been researched concerning forgiveness and reconciliation in palliative care can be placed in the development of the spiritual dimension of palliative care that has taken place over the last two decennia. Two papers from North America give a good overview of the state of the science (Steinhauser et al. 2017; Balboni et al. 2017). For finding a common focus and definition of spiritual care as a dimension of palliative care, the consensus conference organized by Christina Puchalski in 2009 has played an important role (Puchalski et al. 2009). The consensus definition of spirituality that was agreed upon at that conference has inspired the taskforce for spiritual care of the European Association for Palliative Care to adopt a European version that has had a guiding role in many European documents and research projects over the last decade (Nolan et al. 2011).

Looking at both definitions and the way spiritual care has been developed in palliative care, two important observations can be made. First, spirituality is defined as a universal human dimension focusing on meaning, purpose, and connectedness; by doing so the phenomenon is abstracted from its cultural and contextual plurality. The many different and sometimes conflicting understandings of this world and the divine (expressed in cultural and religious textures of meaning that deeply influence and determine our outlook on life) are lost. The good thing about this approach is that it is driven by the sincere intention and ambition to create a space in which people from different religious and cultural traditions can meet each other; are all considered as equal and collaborate to make the best spiritual care possible for patients and families from all walks of life. The problematic side of this is that this abstraction of spirituality is a way of proceeding that fits better with one tradition than another. The concepts and categories of speaking and thinking about spirituality are those of the Western (predominantly North American) mind; they fit better with Christianity than other religious and non-religious traditions.

A second observation is that spiritual care is seen as a dimension of palliative care. Although palliative care is an approach that seeks to be multidimensional and multidisciplinary, in the end, it is the world of medicine that is the standard and framework of reference in terms of what counts as good and reliable knowledge. As palliative care is aimed at quality of life, the religious and the spiritual are seen as supporting quality of life—not the other way around. This has far-reaching consequences for the way forgiveness and reconciliation are discussed and researched. For now, it is important to notice that the development of medical science is one of the great achievements of Modernity—with all its

good and less fortunate sides. As the German sociologist Hartmut Rosa shows in this analysis of the age of acceleration we live in, Modernity has made the world "available", which can be explicated in four elements: Making the world visible, accessible, controllable, and usable (Rosa 2016, 2019). As a result, we have gradually been caught in a reductionist worldview that does not know how to deal with what is "non-available" (*unverfügbar*). Religious traditions, in their best forms, are sophisticated cultural expressions of relating to what is beyond human comprehension and control, and cannot be instrumentalized. As we will see, this is precisely what seems to be missing in contemporary research on forgiveness and reconciliation in palliative care.

## 3. Forgiveness and Reconciliation in Palliative Care: What Has Been Researched So Far?

It is not my intention to give a comprehensive overview of what has been published on forgiveness and reconciliation in the field of palliative care. My purpose is to address an observation that seems to be problematic as a tendency in contemporary research on both themes in palliative care. For that purpose, I will limit myself to discussing some of the more prominent studies that immediately pop up when doing a quick scan in PubMed combining the terms "palliative care" OR "end-of-life" OR "dying" AND "forgiveness". Those who would conduct a comprehensive literature review—which, to date, to my knowledge, is not available (Silva et al. 2017)—will probably find exceptions to the tendency I address. Possible exceptions, however, do not refute the point I want to make.

In 2005, Baker (2005) published a paper on the therapeutic value of a psychosocial intervention in end-of-life care. The paper stated the importance of addressing psychosocial concerns with dying patients as pivotal to facilitating peaceful closure in end-of-life care. In order to facilitate forgiveness and peaceful closure, the literature in this area was examined, and a case example was given of a hospice patient's need for closure and the responsive social work intervention for the patient in his moment of death. In the light of what has been said before, we see how forgiveness is instrumentalized as a way to reach an overarching goal: peaceful closure. The intervention that is described is aimed at controlling this process.

In 2009, Hansen et al. (2009) reported a palliative care intervention in forgiveness therapy for elderly terminally ill cancer patients. They studied the effectiveness of a four-week forgiveness therapy program in improving the quality of life of these patients. All participants completed instruments measuring forgiveness, hope, quality of life, and anger at in a pre-test, post-test 1, and post-test 2. According to their measurement, the forgiveness therapy group showed greater improvement than the control group. Again, forgiveness is instrumentalized in order to improve quality of life. This time, a successful intervention seemed to prove that forgiveness can be managed.

Guilt and forgiveness have been developed into psychological constructs that can be measured. In 2012, Van Laarhoven et al. (2012) compared attitudes of guilt and forgiveness in cancer patients without evidence of disease and advanced cancer patients in a palliative care setting. A total of 97 patients without evidence of disease and 55 advanced cancer patients filled out the Dutch Guilt Measurement Instrument and the Forgiveness of Others Scale. According to the scales, an attitude of nonreligious guilt and forgiveness was found in cancer patients, irrespective of the stage of disease. Interestingly, religious characteristics were significantly associated with attitudes of guilt and forgiveness. One of the advantages of psychological constructs is that they enable measurement, and measurement fits well in the medical regime and the ideals of Modernity. The question, however, is to what extent measuring a construct simultaneously might block the way of understanding the phenomenon that the construct is based upon. An attitude of guilt—just as feelings of guilt—is a psychological category that is just a part of the phenomenon of guilt, which, from a spiritual or religious perspective, is first and foremost a moral category.

Not all interventions in which forgiveness and reconciliation play a role explicitly express this in their name. In 2008, Steinhauser et al. (2008) published a pilot RCT (Randomized Control Trial) study on the question of whether preparation and life-completion discussions improve functioning and quality of life in seriously ill patients. In their conceptual model, they aimed at QoL (Quality of Life),

among other things, and formulated the developmental tasks of expressing regret and forgiveness and accepting gratitude and appreciation. The Outlook tool that they developed consisted of three sessions, the second of which was dedicated to forgiveness. Although the three-armed RCT included 82 hospice-eligible patients and reported improvements in functional status, anxiety, depression, and preparation for end of life, the sample size was not big enough for statistical significance. The Outlook tool has also been tested in its use by nurses (Keall et al. 2011, 2013) and chaplains (Steinhauser et al. 2016). Studies report that the intervention is appreciated by those who are involved (both caregivers and care receivers).

A study that differs from the intervention studies is the one published by Ferrell et al., in which 339 nurses from courses throughout the U.S. and Belize, India, the Philippines, and Romania attending palliative care educational programs shared narratives of their experiences in caring for patients who expressed regret or the need for forgiveness (Ferrell et al. 2014). A total of 346 stories were collected and analyzed using content analysis. Only a few nurses described experiences of witnessing skilled colleagues address forgiveness. Frequently, nurses seemed to offer a "quick fix" (using expressions like "It's ok" or "I'm sure he will forgive you"). Many nurses described occurrences of "peaceful deaths" following acts of forgiveness and reconciliation as "miraculous". It would be interesting to compare these findings with Rosa's analysis of "unavailability" as something that is problematic to the modern scientific mind. Perhaps the word "miraculous" expresses precisely the experience of forgiveness and reconciliation as something alien to the world of nursing and medicine because it cannot be managed.

Research among patients and caregivers in relation to forgiveness and reconciliation shows that both themes are not easy to handle for caregivers (Leget et al. 2013; Wittenberg et al. 2016), but patients receiving palliative care also seem to wish to live for the present with as much normality as possible and show only minor concern for their past and future, including being reconciled and feeling forgiven (Vilalta et al. 2014). One explanation for these findings might be that, next to the dimension of "unavailability", forgiveness and reconciliation are also related to feelings of shame or cultural taboos. In that respect, next to the issue of effectiveness, the interventions could be seen as practices in which themes that are difficult to deal with find a new ritual form, which is no longer embedded in a religious context, but now in a secularized and medical context.

A few observations are interesting to conclude with. Firstly, there is more research on the role of nurses than on the needs of patients. There seems to be no research on the role of physicians. Secondly, although there seem to be a few tools available, much of the research about forgiveness might be under the radar because it is part of life review, QoL, or other psycho-social and spiritual concepts and approaches. Thirdly, and more importantly, our quick scan has demonstrated that the research done so far implicitly focuses on the psychological dimension of forgiveness and reconciliation, and does not address the moral dimension of the phenomena and the aspect of "unavailability" connected with it.

## 4. Forgiveness and Reconciliation between Psychology and Ethics: Some Questions

One of the most influential and successful researchers on forgiveness in health care is clinical psychologist Everett Worthington. In the title of a seminal paper—in which his theory and research hypotheses are proposed—forgiveness is defined as an emotion-focused coping strategy that can reduce health risks and promote health resilience (Worthington and Scherer 2004). The paper is interesting because it may help to get a clearer picture of the way the phenomenon of forgiveness is approached according to the logic of Modernity.

Worthington et al. distinguish between decisional and emotional forgiveness. Decisional forgiveness is defined as the behavioral intention to resist an unforgiving stance and to respond differently to a transgressor. Since behavioral intentions are within our control as free human actors, this type of forgiveness is based on an intentional decision: an act within our control as human beings. Emotional forgiveness, on the other hand, is defined as the replacement of negative unforgiving emotions with positive other-oriented emotions. Again, the language is that of intentionality and control: Emotions can be replaced just like furniture in a room.

The reason why unforgiveness should be avoided, according to Worthington et al., is connected with the central value of health. The line of reasoning is as follows: Unforgiveness is stressful. This stress can be avoided through coping mechanisms and through forgiveness. Thus, forgiveness can be deployed as a coping strategy to reduce the stress of unforgiveness. This strategy is positively correlated with health. With this goal in mind, this paper sketches a research agenda aimed at better understanding (1) whether forgiveness affects health, (2) how it might do so, and (3) how interventions might be crafted.

From the perspective of healthcare, the logic of Worthington's paper seems clear and convincing. However, what happens if we shift to the perspective of spirituality and religious traditions? Although spirituality and religion are not (or should not) be a threat to a healthy lifestyle—research from the United States of America rather suggests that the moderation associated with specific religious lifestyles is rather positively correlated with good health and a longer life span—health is not the most important thing for people who have a religious mission. Let us take the example of three people who have led meaningful lives and whose spiritual, intellectual, and social legacies is beyond discussion. Francis of Assisi, founder of the Franciscan order, died at the age of 44, exhausted and in a bad physical condition. Philosopher, religious thinker, and social activist Simone Weil died from exhaustion at the age of 34. Martin Luther King, Jr. was killed at the age of 39.

Clearly, to these people, social justice was far more important than living a long and healthy life. For them and many who admire them, the quality of their lives was defined by moral and spiritual standards rather than medical or psychological categories. With (social) justice, we enter the domain of morality or ethics. In this domain, there is a different hierarchy of values from that in the domain of healthcare and psychology, as well as a different tradition of understanding and researching reality. Since palliative care aims to promote quality of life in all its dimensions, this approach is inherently open to various important values that may be in conflict with each other. From a moral perspective, however, forgiveness and reconciliation seem less manageable and are not subordinated to health. The picture becomes more complex if one undertakes a philosophical analysis of forgiveness and reconciliation. For this, I will turn to the German philosopher Svenja Flaßpöhler, who has written a thought-provoking book on guilt in which she brings together the thoughts of important 20th century philosophers on the subject (Flaßpöhler 2017).

## 5. Guilt: Excusing (*Verzeihen*) or Forgiving (*Vergeben*)?

Embedded in an autobiographical context, Flaßpöhler makes an interesting distinction between two German words for forgiving (Flaßpöhler 2017, pp. 20–23). The first word is "*verzeihen*" and could be translated as the English word "excusing": It refers to the act of letting go of guilt and refraining or holding back from anger, hate, and retribution. Excusing is clearly something within human possibilities, a free moral act that deviates from the order of justice. Instead of "an eye for an eye, a tooth for a tooth", or an economic rationality of *do ut des* (I give so you will give in return), excusing is something that is done after a decision that someone is exempted from paying back his or her guilt. Excusing is something that is easily done in everyday situations and helps to smoothen human relationships and to relax the atmosphere between human beings.

The other German word that Flaßpöhler discusses is "*vergeben*", which can be translated as "forgiving" or "pardoning". This word has religious connotations, and is used in the context of the relationship between human beings and the divine. Etymologically, in many languages, this word contains the concept of giving: forgiving (English), *vergeben* (German), *pardonner* (French). What distinguishes forgiving from excusing is that, in this case, the moral act is not passive (refraining, holding back), but active (giving). Forgiving also breaks the order of justice because something is given beyond the order of justice.

What unites excusing and forgiving is that both break with linear causality and calculation. Both concepts contain a core of non-availability—something connected with the creative freedom that,

in the Christian tradition, is related to the concept of "grace". A free creative act cannot be planned or managed; it happens as a free act and a gift at the same time.

Having introduced this distinction, Flaßpöhler uses the word *verzeihen* because she reflects on the way human beings deal with guilt amongst each other. She is less interested in the relationship between human beings and the divine. In what follows, I will use the English word "forgiving" for the sake of clarity because, in English, both *verzeihen* and *vergeben* are usually translated into the more common word forgiving. Flaßpöhler investigates to what degree we can understand this (non)act of forgiving beyond the logic of retribution. She develops the moral perspective on forgiving by asking whether this act is a way of understanding (Flaßpöhler 2017, pp. 37–81), loving (pp. 83–126), or forgetting (pp. 127–95).

*5.1. Does Forgiving Mean: Understanding?*

One of the ways in which people cope with injustice being done to them is by trying to understand what has happened. Injustice can be seen as something that deviates from a certain order of things. Literally speaking, the world is not "in order" anymore, and this creates a feeling of uneasiness. When bad things happen to good people, what confidence can we have in the world we live in? An explanation in terms of logic or causal relations can help to restore the order again, and this may lead to a renewed sense of confidence and belonging. So, it seems that understanding injustice might be a first step towards excusing what has happened in order to let go of it and continue with one's life.

Trying to understand something evil also has a second function: It helps to combat powerlessness by actually doing something and taking back control over one's life. Injustice or evil have the power to dominate one's life when what has happened seems to be frozen in the past, and one is paralyzed by the thought that it can never be made undone. Understanding might be a way of taking control again—putting the evil into a new perspective and avoiding demonization.

Flaßpöhler's illustrates this with the case of Gisela Mayer, the mother of one of the children who was killed during a high school shooting in Minnenden, Germany on 11 March 2009. In a conversation with the author, this mother tells about her long process of forgiveness. In the beginning, she was so hurt by what had happened that she had to take distance from the events. She was not able not speak out the name of the offender, and looked upon the high school killings as a non-human act, similar to a natural disaster. After this phase of impotence and being frozen, she discovered a huge anger inside her. From the moment she was able to speak out the name of the offender, however, she was able to recognize that the killing had been a human action after all. Then, she started to ask herself how it could have been possible that a 17-year-old boy had killed 15 people before killing himself. She learned that the boy must have been very isolated and lonely, suffering from a dominant father who had made him feel that, as a son, he would never be able to live up to his father's expectations. The boy had overcompensated with extremely violent behavior, especially, it seems, against women, since almost all victims were girls and women.

By understanding the event from the offender's perspective, the mother was able to give back the 17-year-old boy his humanity, not reducing him to the last fatal moments of his life. In a similar way, she was able to give back her daughter a life worth remembering, not reducing her to her last moments of being a victim that was killed by a fellow student. From a psychological perspective, all this makes sense, and is probably a sound way of coping with something that is hardly possible to live with and has the capability of ruining one's life. From a moral perspective, however, it is clear that there are some problems.

In the first place: Understanding does something with framing the harm that has been done, but it can never take away the damage that has been done or cause the evil to be undone. Understanding refers to having insight into some kind of causal relation, but this insight is something different from forgiving. Understanding does not take away guilt. Moreover, if insight into causal relations would have the power of excusing or forgiving, it would mean that the idea of morality as a whole would become impossible. Morality is based on the idea that human beings have free will. Based on this

idea of free will, human beings can take responsibility and be held accountable. Denying the free will would mean that our entire system of justice would become impossible. So, the question remains: Is real forgiveness not exactly beyond understanding, beyond a rationality of retribution and justice? Is real forgiveness not rather a gift than an exchange? Could it be that we understand more of forgiving when we see it as a gift of love?

*5.2. Does Forgiving Mean: Loving?*

At first sight, equating forgiving with loving seems a more promising move than seeing forgiving as a way of understanding. Love is basically a free gift that cannot be demanded according to an order of justice. The distribution of affect follows a different rationality from that of need or desert (Walzer 1983, pp. 227–42). Love does not seem to be distributed according to a linear causality or clear logic. This characteristic of unavailability, which is similar to both forgiveness and love, seems to suggest that forgiveness might essentially be an act of love.

On the other hand, both love and forgiveness do not seem to be completely characterized by irrationality and unavailability. As the French philosopher Jacques Derrida points out, there is a fundamental and indissoluble tension between what he calls the utopian creative (ex nihilo), non-economical, and unconditional ideal of forgiveness on the one hand, and the historical, conditional, and economical act of forgiving on the other. It is almost impossible to forgive someone who does not show any remorse, like it is almost superhuman to love without being loved in return. So both love and forgiveness seem to follow a certain economy, although their distribution cannot be reduced to this economy.

Although love and forgiveness may be connected, Flaßpöhler warns us against equating them too easily by telling the story of a man who is in prison because he has killed a woman who he considers to be the love of his life. Although justice has been done and the convict is paying for his crime with many years of imprisonment—and even though the chaplain tells him that God has forgiven him because he has shown repentance—the man refuses to be forgiven. Not even God can forgive him, he states, because he has killed the love of his life. Forgiveness for him would take away the last connection with the woman he has loved above all. Because he still loves her, he refuses to be forgiven. His guilt keeps him alive by connecting him with this great love. Clearly, in this example, forgiving can never be loving because it would take away the love for this woman. To the contrary: In this case, the act of loving is incompatible with the act of forgiving.

One way to solve this dilemma is proposed by the French philosopher Emmanuel Levinas. According to his view, the creative act of love cannot reverse time and make the past undone. Forgiveness, however, can make it *as if* a deed had not been done. By forgiving, the forgiver has a gift to the offender: He gives the offender a new, forgiven past. In this way, the offender is released from what has happened, and this allows for a new beginning. The future is reopened for the offender to begin again *as if* the deed was never done.

Although this creative act of forgiveness may be an effect or even an expression of love, the question is whether forgiveness as such can be seen as an act of love. By rewriting the past, we seem to move towards the third option that Flaßpöhler discusses: The question of whether forgiving can be seen as a way of forgetting.

*5.3. Does Excusing Mean: Forgetting?*

As the argument of Levinas shows, forgetting is not the same as amnesia. In fact, forgetting as it is used in Levinas' account of forgiveness is an act by which the psychological meaning of the past changes. This act, however, seems to be more similar to a long process than a moment in time—just like the process Gisela Mayer experienced as she gradually understood the circumstances that contributed to a 17-year-old boy killing her daughter.

Following this idea of forgiving as a way of rewriting the past, from a psychological point of view, we might seem to be following a promising lead. From a moral point of view, however, the question

might come up as to whether there is a limit to such forms of forgetting as a way of forgiving, for who is entitled to forgive whom?

During one of the moral case deliberations I chaired during a palliative care education program in Germany, one of the nurses came up with a case of an older patient who refused to take off his shirt in order to be washed. After a period of building confidence and talking, he finally gave in and showed one of the nurses why he had refused to bare the upper part of his body. One side of his body was covered with a huge tattoo of a swastika. The man was so ashamed of this remnant of earlier days that he did not know how to cope with it. Heavily shocked by this event, as a consequence, many nurses refused to take care of the man any further. How should this be evaluated?

On the one hand—and considered from the perspective of professional ethics—this case might seem to be relatively easy to deal with. Patients have a right to be taken care of, regardless of their walk of life or personal history. On the other hand, caregivers are human beings with their own personal histories and emotions. Even if one would forgive the older man for his foolish behavior in the past, would an act of forgiveness not at the same time imply an offense to the victims of the Nazi regime he seems to have endorsed?

Flaßpöhler reflects on the limits of forgiving by using a very controversial example—the story of Eva Mozes Kor, a woman who had lost both of her parents and two older sisters in Auschwitz. She herself, together with her twin sister, Miriam, had been a victim of Josef Mengele's medical experiments. In 1995, at the 50th celebration of the liberation from the Nazi regime, Eva Mozes Kor read a statement in Auschwitz, in which she "gives amnesty to the Nazis who have been involved directly or indirectly in the killing of my family and millions of other people" (Flaßpöhler 2017, p. 155).

Who—if anyone at all—could be more entitled to proclaim such an act of forgiveness than a woman who survived the atrocities she refers to herself? Yet, it raises the question on whose behalf she is speaking and whether it is appropriate to extend one's forgiveness beyond what she has undergone herself. Is it possible to forgive perpetrators whose victims cannot speak for themselves anymore? If forgiving means forgetting, would this not imply that some deeds may never be forgiven because they may never be forgotten?

According to the French philosopher Vladimir Jankélévitch, the ontological evil of the Shoah cannot be forgiven because what happened there was against humanity. The crimes that have been committed in Auschwitz cannot be paid back; they are without limitation, and time has no influence on them. The reason for this, according to Jankélévitch, is that time belongs to the natural order, but crimes belong to the order of morality. Offenses against humanity are to be remembered for the sake of upholding our standards of humanity and out of respect for those whose existence has been denied and erased. Forgiving the Nazis would mean to wipe out the names of their victims. What might seem to be promoting mental health from a psychological perspective seems to be unacceptable from a moral perspective.

## 6. Concluding Remarks

I opened this paper with the observation that forgiveness is an important theme in end-of-life care in all spiritual and religious traditions, albeit framed differently. Looking at recent research on forgiveness in palliative care literature, we discovered that this is predominantly carried out from a psychological perspective. According to this approach, forgiveness is seen as something that can be managed and taught in order to reduce stress and promote health. There is no doubt that this approach has its merits and is useful for dealing with guilt from the individual perspective of one's own psychological health. On the other hand, we have learned from Svenja Flaßpöhler's reflections that, from a moral perspective, the experience of feeling guilty and suffering from the stress that this brings is something completely different from guilt between two parties. Where the psychological approach of feeling guilty stays within the logic of Modernity that helps to manage one's relationship with one's feelings, the moral and religious concept of guilt confronts us with the limits of Modernity. Here, we encounter a holding-back or giving beyond the logic of calculative justice. From a moral perspective,

guilt and forgiveness confront us with values and laws that are beyond our design and control. We are left with a gap between a psychological and a moral approach to forgiveness and reconciliation.

As a conclusion, and considering the research on forgiveness and reconciliation in palliative care, I would propose the following. In the first place, for the sake of scientific rigor, I think that, in the future, it is important to make a clear distinction between the psychological category of feelings of guilt, forgiveness, and reconciliation on the one hand, and the moral and religious phenomena of guilt, forgiveness, and reconciliation on the other. The first category concerns the individual and the relationship with oneself. The second category regards the social and societal dimension of existence—the relationship between individuals who are seen as moral actors. From a moral perspective, guilt, forgiveness, and reconciliation cannot be dealt with without including all those who are involved in one way or another.

Secondly, considering what has been done in the past twenty years in palliative care research, what is needed in the future is the integration of both the psychological and the moral (and religious) approaches to forgiveness. Such an integration might bring more complexity and depth to psychological approaches and more empirical groundedness to philosophical approaches. In order to develop such research, close collaboration is needed between psychologists, philosophers, and scholars who are experts in the fields of spirituality and religious traditions. Bridging the gap asks for efforts from all disciplines.

Thirdly, such a collaboration would require an interdisciplinary methodological discussion of the relationships between different fields or disciplines and the limitations of the forms of knowledge they produce. There is no questioning in palliative care about the importance of multidisciplinary care and research. However, because of the way we have organized our universities, education programs, and research activities, research following the medical model and the logic of Modernity has more status and impact than philosophical reflection. This is a bias and fault that can be corrected and repaired. Avoiding one-sidedness can be managed and taught, just like the psychological concept of forgiveness. Not doing so is more than a scientific limitation: It produces and spreads reductionist accounts of forgiveness and reconciliation that are not helpful for those who are confronted with guilt.

**Funding:** This research received no external funding.

**Conflicts of Interest:** The author declares no conflict of interest.

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
