# Peer review of "Forgiveness and Reconciliation in Palliative Care: The Gap between the Psychological and Moral Approaches"

_religions, doi:10.3390/rel11090440_

Round 1

Reviewer 1 Report

On almost every page there are places that need a proof-reader who can correct mistakes of grammar, punctuation and incomplete thoughts. Most of these errors are easily corrected by someone who reviews the technical aspects of the writing. The theme of the essay is worthwhile. The one caution I would make is to be very careful about introducing the acts of forgiveness embodied by the survivors of the Holocaust. That atrocity is on a scale that would not be applicable to cases dealing with people in palliative care, working through reconciliation on a person-to-person level.

But this study has true merit.

Author Response

Thank you for your review and helpful comments. I apologize for the many mistakes in the previous version, which were due to pre-vacation and deadline stress. I performed an extensive editing of English language and style. All your suggested amendments have been inserted in the new manuscript. There I also added a passage relating to my personal experience of how in Germany professionals in palliative care from time to time still struggle with the Holocaust, also on a person-to-person level, in order to make a better bridge to the text of Flaßpöhler.

Reviewer 2 Report

This paper need some corrections, but honestly could benefit from a re-write that would work on greater focus and to settle on just what the author really wishes to prioritize.As written we get several different sections that could be better focused.  We have 1) a literature review; 2) a reference to the relation to palliative care and forgiveness; 3) a mini-treatise on spirituality from a cross-cultural perspective; and 4) a synopsis of the work of Svenja Flasspoehler.  A stronger article would give pre-eminence to one of these 4 and thereby shorten the other 3 and integrate them into service to the chosen priority.  This would likely change this piece from an "average" article to one of "high" rating.

The items that should (must) be changed:

Line 179 change "Much" in the middle of the sentence to "much" [of the research] [this looks like a sentence that was probably changed in the course of the writing, and as currently stands is incorrect in English grammar.

Line 195 change "resist and unforgiving stance" to "resist an unforgiving stance..."

Line 217, factual inaccuracy.  "Martin Luther King" died at age 84 in 1984. He was the father of his better known son, who almost always in the USA is referred in academic writing at "Martin Luther King, Jr." (who died in 1968 at age 39).

Author Response

Thank you for your review and helpful comments. I apologize for the many mistakes in the previous version, which were due to pre-vacation and deadline stress. I performed an extensive editing of English language and style. All your amendments have been inserted in the new version.

I have considered your suggestion of rewriting the paper and focus on one of the four parts you identified. However, given the fact that my intention is to identify a gap between two disciplines and two ways of approaching forgiveness and reconciliation, I did not manage to make a better version of the paper leaving out one of the four parts. In order to convince the reader that there is a problem in palliative care research, I need the evidence presented in the quick scan review; and in order to describe the gap, I need to both reflect on the psychological and the philosophical approach to forgiveness and reconciliation. I am aware of the fact that this approach might be unsatisfying for readers from both diciplines as the paper deviates from what is considered to be a standard paper in their discipline, but I am afraid this is ineviable in the kind of interdisciplinary approach I intend to follow.

I rewrote crucial parts of the text in order to clarify the composition of the paper and improve its consistency. I hope this will take away your concerns.